# Conservation Genetics of the Critically Endangered Southern River Terrapin (*Batagur affinis*) in Malaysia: Genetic Diversity and Novel Subspecies Distribution Ranges

**DOI:** 10.3390/biology12040520

**Published:** 2023-03-29

**Authors:** Mohd Hairul Mohd Salleh, Yuzine Esa, Suriyanti-Su Nyun Pau

**Affiliations:** 1Department of Aquaculture, Faculty of Agriculture, Universiti Putra Malaysia, Serdang 43400, Selangor, Malaysia; 2Royal Malaysian Customs Department, Persiaran Perdana, Presint 2, Putrajaya 62596, Putrajaya, Malaysia; 3International Institute of Aquaculture and Aquatic Sciences, Universiti Putra Malaysia, Port Dickson 70150, Negeri Sembilan, Malaysia; 4Department of Earth Sciences and Environment, Faculty of Science and Technology, Universiti Kebangsaan Malaysia, Bangi 43600, Selangor, Malaysia; 5Marine Ecosystem Research Centre (EKOMAR), Faculty of Science and Technology, Universiti Kebangsaan Malaysia, Bangi 43600, Selangor, Malaysia

**Keywords:** population genetics, malaysia, phylogenetic, neutrality test, demography history, genetic diversity, terrapins

## Abstract

**Simple Summary:**

A novel comprehensive population genetics study of two *Batagur affinis* subspecies in Malaysia is part of a conservation genetics effort aimed at long-term sustainability. As the conservation status of this species is critically endangered, we found only six different haplotypes from four study areas and low genetic diversity using D-loop markers. Nevertheless, using genetic markers, we revealed the presence of one of the Southern River terrapin subspecies (*B. affinis edwardmolli*) in Kedah, Malaysia.

**Abstract:**

A population genetics study was carried out on the Southern River terrapin (*Batagur affinis*) from four places in Peninsular Malaysia: Pasir Gajah, Kemaman (KE), Terengganu; Bukit Pinang (BP), Kedah; Bota Kanan (BK), Perak; and Bukit Paloh, Kuala Berang (KB), Terengganu. The goal of this study is to identify genetic differences in two subspecies of *B. affinis* in Malaysia. No previous reports were available on the genetic diversity, phylogenetic relationships and matrilineal hereditary structure of these terrapin populations in Malaysia. The sequencing identified 46 single nucleotide polymorphisms that defined six mitochondrial haplotypes in the Southern River terrapins. Tajima’s D test and Fu’s Fs neutrality tests were performed to evaluate the signatures of recent historical demographic events. Based on the tests, the *B. affinis edwardmolli* was newly subspecies identified in the west coast–northern region of Kedah state. In addition, the *B. affinis edwardmolli* in Bukit Paloh, Kuala Berang (KB), Terengganu (Population 4), was shown to have a single maternal lineage compared to other populations. Low genetic diversity, but significant genetic differences, were detected among the studied Southern River terrapin populations.

## 1. Introduction

The Southern River terrapin (*Batagur affinis*) [1] (also known locally as Tuntung Sungai) is a freshwater turtle found in large rivers all over the Indochina region. *B. affinis* (Figure 1) is exclusively distributed in Indochina, including Sumatra, Singapore, Peninsular Malaysia, Thailand, Vietnam, and Cambodia [2,3]. This species is listed by the Turtle Conservation Coalition [4] as one of the world’s 25 most endangered freshwater turtle species. In addition, it is classified as “critically endangered” (CR) on the International Union for Conservation of Nature’s (IUCN) Red List [2] due to several threats, including sand mining, egg consumption, and water sedimentation [5].

*Batagur affinis* is one of 24 turtle species found in Malaysia [5]. Currently, Malaysia is the only place on the planet where *B. affinis affinis* is still found in its natural habitat [7]. Wild *B. affinis* populations in Peninsular Malaysia have been documented based on minor morphological differences and coloration with collections of three nuclear DNA markers and three mitochondrial markers that are used to differentiate between two subspecies of *B. affinis* populations in Peninsular Malaysia [8]. The Western coast populations are placed mainly in *B. a. affinis*, while the Eastern coast populations are assigned to *B. a. edwardmollis* [2]. The nominate subspecies, *B. affinis affinis,* is now only found on the western coast of Peninsular Malaysia, as it is believed to be extinct in the wild in Thailand [2] and Sumatra, Indonesia [9].

On the other hand, *Batagur affinis edwardmolli* was once found from Singapore to Indochina. It is now thought to be extinct in the wild, at least in Thailand, Singapore, and Vietnam [2]. This means that the only genetic stock of Indochina’s Southern River terrapin remains in Malaysia and Cambodia.

To sustain the genetic stock, molecular methods are often used to better understand the phylogenetic relationships between different populations [10]. DNA sequence analysis can be used to determine evolutionary relationships, degrees of variation, and geography. Substructures within and between populations [11,12] can be compared by using mitochondrial deoxyribonucleic acid (mtDNA) molecules, which have more copies than nuclear DNA (nDNA) molecules [13].

The D-loop is an essential transcription regulator found in mitochondrial genomes. It has been reported that this region has more sequence variation than other regions of mtDNA sequences [14]. Variation in the D-loop region has been used to determine the origin and diversity of many breeds of domestic animals [15]. Further, mtDNA polymorphisms have remained relevant to the study of population structure and intraspecific variation [16,17,18,19,20,21].

Due to the current CR status of *Batagur affinis*, genetic factors can further and significantly impact the population’s risk of extinction. The loss of genetic diversity can result in the reduction of long-term evolutionary potential [22,23]. A common extinction mechanism, known as inbreeding depression, reduces populations’ reproductive success and survival rate, making it one of the most common and potent extinction mechanisms [22,23]. The loss of genetic diversity in a greatly diminished population can hasten the extinction of the population in question [22]. As wild *B. affinis* populations have declined to critically low levels in recent years, in-situ conservation programmes and captive population reintroductions were implemented as the most widely used methods of protecting endangered species [24]. The primary objectives of captive founder compilation and captive breeding programmes are to produce self-sustaining captive stock with the closest behavioural and genetic similarities to their endangered wild counterparts possible. Most captive breeding programmes (ex-situ) are designed to achieve this goal [25,26,27,28,29].

Previously, no research has been conducted on Southern River terrapins using mtDNA D-loop sequence variability in Malaysia. Therefore, this study is crucial, as we are the first to investigate the null hypothesis stating that lower genetic variability is associated with genetic differentiation between two subspecies. In contrast, the alternative hypothesis states that higher genetic variability means no genetic difference between two subspecies. Additionally, this study was valuable for evaluating the present genetic variation between two subspecies of *Batagur affinis* in Malaysia. As a result, this conservative genetics analysis will aid decision makers in the conservation, utilisation, and exploitation of *Batagur affinis* in Indochina.

## 2. Materials and Methods

### 2.1. Sample Collection

The sample for this study was made up of 120 *Batagur affinis* samples from four populations on the east and west coasts of Peninsula Malaysia (Figure 2). The study sites are Pasir Gajah, Kemaman (KE), Terengganu (4.2524° N, 103.2957° E); Bukit Pinang (BP), Kepala Batas, Kedah (4.2221° N, 100.4370° E); Bota Kanan (BK), Bota, Perak (4.3489° N, 100.8802° E); and Bukit Paloh, Kuala Berang (KB), Terengganu (5.0939° N, 102.7821° E). A total of 30 *B. affinis* individuals were sampled at each location. Following the protocols in [30], blood was drawn using two different venipuncture techniques, including the subcarapacial venous plexus (SVP) and the jugular vein. Around 1.5 mL of blood was collected into a 2 mL microcentrifuge tube and mixed with 0.5 mL of EDTA (1:3 ratio) for preservation before being stored at −20 °C. The Department of Wildlife and Parks, Peninsular Malaysia, approved B-00335-16-20 for a research and field permit. 

### 2.2. DNA Extraction and PCR Amplification

Nucleic acids were isolated from 200 µL of each EDTA whole blood sample. Following cell lysis and protein denaturation, extractions were carried out using an automated system, the ReliaPrep^TM^ Blood gDNA Miniprep System (Promega, Madison, WI, USA), with Binding Column technology. The input volume of the EDTA whole blood sample yielded a final extraction volume of 200 µL.

The amounts of extracted DNA were determined using a Thermo Scientific™ NanoDrop 2000c spectrophotometer (Thermo Fisher Scientific, Waltham, MA, USA). The automated system’s competence to extract nucleic acids of great purity was verified by direct gel loading. Following NanoDrop quantification of the isolated nucleic acids, the results were loaded directly into the 1% (*w*/*v*) agarose gel with molecular markers.

A primer set of 5′-TTTTTCCCCTAGCATATCACCA-3′ (forward) and 5′-AGTTGCTCTCGGATTTAGGG-3′ (reverse) designed previously by [31] was used. PCR amplification for gene D-loop fragments was performed in a Go Taq Flexi PCR (Promega, Madison, WI, USA) reaction mixture containing 2 µL DNA template, 0.5 µL primer, 5 µL 5× PCR buffer, 2 µL × 25 mM MgCl_2_, 0.5 µL dNTP, 0.2 µL Taq DNA polymerase, and 14.3 µL double-distilled water (ddH_2_O). Denaturation at 94 °C for 3 min was followed by 25 cycles of denaturation at 94 °C for 35 s. Meanwhile, the primer annealing stage was completed at 60 °C for 1 min and 30 s, followed by a 2 min extension at 72 °C and a final 2 min extension at 72 °C. The purified PCR products were shipped to a private company (First BASE Laboratories Sdn Bhd, Seri Kembangan, Selangor, Malaysia) for D-loop sequencing with only the forward primer. The reverse primer was then used in a sequencing procedure on selected samples (haplotype (HAP)) to confirm the polymorphism in the mtDNA sequence with the forward primer. Finally, mtDNA D-loop sequences from different haplotypes were then sent to GenBank under the numbers MZ555651–MZ555656.

### 2.3. Data Analysis

#### 2.3.1. General Characterisation and Population Diversity

The ClustalW algorithm in MEGA X (Global SaaS Software Company, Paris, France) [32] was used for multiple sequence alignment, and DnaSP 6.12.03 (Universitat de Barcelona, Spain) [33] was used to summarise the haplotype distribution of the collected data as well as to identify the polymorphic sites (PS) or Single Nucleotide Polymorphisms (SNPs) [34]. Arlequin 3.5 was used to determine the haplotype diversity (Hd) and nucleotide diversity (π) at the population level [35]. 

#### 2.3.2. Population Structure and Demographic History

The Analysis of Molecular Variance (AMOVA) [36,37] was used to assess genetic diversity at different hierarchical levels. This analysis was performed using Arlequin 3.5 (UBC, Canada) [35]. Arlequin [38] was also used with 1000 random permutations to assess hierarchical genetic structure with conventional *F_ST_* (*cF_ST_*) values in AMOVA and Mantel tests [39], including the estimation of pairwise Tamura–Nei genetic distances, Pairwise Fixation Index (*F_ST_*), and the probability (Chi-square test (χ² test)) [40] for all the population pairs. The pairwise fixation index (*F_ST_*) and pairwise Tamura–Nei results are needed to assess the degree of geographic structuring of genetic variability. Then, we used DAMBE 6 (University of Ottawa, Ottawa, ON, Canada) [41] to test for saturation across all codons. 

A Spatial Analysis of Molecular Variance (SAMOVA) was performed in SAMOVA 2.0 (University of Bern, Switzerland) to find the groups of populations that are spatially homogeneous and maximally dissimilar from each other and confirm the genetic barriers between these groups [42]. The highest within-group (*F_CT_*) variance determined the best number of groups, k (the maximum was k = 3). To account for the family-wise error rate of = 0.05, the False Discovery Rate procedure was used [43].

Two statistical tests, Tajima’s D [44,45] and Fu’s FS [46], were conducted to look for population growth in the past. As a first step, statistical analyses were carried out using the software Arlequin 3.5, and *p*-values were calculated using 1000 simulations under the selected neutrality model. Using the DnaSP 6.12.03 software, more mismatch frequency graphs were made to show the difference between nucleotide site differences and those that stayed the same over time [44,45]. Second, Harpending’s Raggedness Index (RI) [47] was used to determine how the population was changing. The goodness of fit tests for a model of population expansion were used [48] to measure the Sum of Squared Deviations (SSD) [49] between the observed and expected deviations for each population in Arlequin 3.5. This metric measures how smooth the distribution of mismatches is. A result that is not significant means that the population is growing faster than the distribution of mismatches [47]. In the end, a parametric bootstrap method was used to test the spatial expansion hypothesis with 1000 copies of the RI and SSD. 

#### 2.3.3. Phylogenetic Relationship

The tree was made using the Maximum Likelihood (ML) [50] and Maximum Parsimony methods [51], with a confidence level estimated from 1000 bootstrap replicates (Appendix A) to compare with the Bayesian analysis described below.

The best-fitting evolution model for each sequence studied was estimated by employing the Akaike information criterion (AIC), with a correction for sample size implemented using jModelTest2 on XSEDE 2.1.6 [52]. In the phylogenetic analyses, the best models of sequence evolution found by jModelTest2 for both coding and non-coding sequences were used.

*Batagur affinis affinis* and *B. a. edwardmolli* mtDNA D-loop sequences were aligned using ClustalW in MEGAX [32] software. Then, Maximum Likelihood (ML) [53] analyses were conducted. All the sequences generated multiple sequence alignments with the same length and starting point shown in the sequences.

The phylogenetic reconstruction was performed by using the IQ-tree [54] on XSEDE [55] through the online CIPRES Science Gateway V.3.3 (https://www.phylo.org/portal2/login!input.action; accessed on 14 May 2021) [56]. The trees obtained were visualised in FigTree v1.4.4 [57]. Firstly, the phylogenetic tree topology and divergence times were estimated jointly using the BEAST v2.6.6 package [58,59]. 

The BEAUti 2, [60] program was performed to unlink the substitution models of the data partitions and to implement the models of sequence evolution identified as optimal by jModelTest2. The “Clock Model” was set as a strict clock with uncorrelated rates, while the “Tree Model” was set to a Yule process of speciation. The sequences were analysed using a relaxed molecular clock model that allows substitution rates to vary across branches based on an uncorrelated lognormal distribution [58] and sets the species tree priors as a Yule Process. 

Using Bayesian Markov Chain Monte Carlo (MCMC) simulations for 100,000,000 generations with a sampling frequency of 5000, two simultaneous analyses were conducted. The nucleotide substitution model for ML was set as TN93 empirical. The bootstrap analysis provided branch support (1000 pseudoreplicates), and all other parameters were set to default values. Finally, FigTree v1.4.4 was used to plot the phylogenetic trees. 

To construct the phylogenetic trees, the mitochondrial sequences of *Batagur affinis edwardmolli* (MN069309) and *Batagur kachuga* (MZ156025) were selected from the GenBank online database as out-groups [31,61]. A Median Joining (MJ) network analysis was performed using NETWORK 10.2 (Fluxus Technology Ltd., Bandelt, Germany) [62].

## 3. Results

### 3.1. Population Diversity

In this study, 120 Southern River terrapin samples were analysed. To evaluate maternal lineage among the populations, these *Batagur affinis* samples were divided into four population regions. From these samples, 30 samples were assigned to each population region. Six distinct D-loop haplotypes were found, each determined by 46 SNPs or substitution numbers inside the 651 bp fragments. These SNPs contained 43 transitions and three transversions at positions 1, 11, and 23, respectively (Table 1). All the substitutions occurred at third codon positions, and this is a common feature of mtDNA [63]. As shown in Appendix A, the [64] (F84) distance was used to plot transitions and transversions against divergence.

The number of haplotypes varies among different population regions, ranging from one to three (Table 2). Among these haplotypes, HAP_2 was the leading one, followed by HAP_3, HAP_5, HAP_6, HAP_4, and HAP_1 (Figure 2). Moreover, Population 2 from BP, Kedah, and Population 3 from BK, Perak, showed high variability among these SNPs, while Population 4 from KB, Terengganu, was shown to have the lowest variability. Haplotype diversity (Hd) (also known as gene diversity) signifies the possibility that two randomly selected alleles are different. In contrast, nucleotide diversity (*π*) is defined as the average number of nucleotide differences per site in pairwise comparisons between DNA sequences [66]. The Hd value ranges from 0 to 0.5356. The π was quite low in all the tested populations, ranging from 0 to 0.0123. The size of some populations and how they have bred in the past may explain why the genetic diversity varies.

### 3.2. Population Structure

In the current analysis (Table 3), within-population variation was measured to be 7.71%, and among-population variation was 92.29%, as indicated by the analysis of molecular variance (AMOVA). The highest *F_CT_* was used to characterise the best population arrangement in SAMOVA with the number of groups (K). However, in this study, SAMOVA demonstrated that *F_CT_* decreased with population partitioning into the most group in all the analyses performed. It should be emphasised that population structure analysis using one or more single population groupings cannot determine group structure [67]. When more than one population was investigated in the study, the optimum population configuration with the highest *F_CT_* was chosen [68]. Based on the SAMOVA, the population arrangement with K = 2 was the best, as characterised by the highest *F_CT_* (0.9439). According to the SAMOVA analysis, the division of four population regions into two groups based on the majority of two subspecies (Group 1 (*Batagur affinis affinis*): Bota Kanan, Perak, and Bukit Pinang, Kedah; Group 2 (*B. affinis edwardmolli*): Kemaman and Kuala Berang, Terengganu) resulted in even higher and significant genetic variations between groups (*F_CT_* = 94.39%, *p* < 0.00001).

Table 4 describes a genetic analysis of the population structure of *Batagur affinis* using Tamura–Nei genetic distances and the fixation index (*F_ST_*). The data is presented in a matrix format with pairwise comparisons of genetic distance (Tamura–Nei) below the diagonal and *F_ST_* values and probabilities of population differentiation (based on a chi-square test with 110 permutations) above the diagonal.

The pairwise values above the diagonal in the matrix indicate the *F_ST_* values between each pair of populations, and the probability of population differentiation based on a chi-square test with 110 permutations. Pairwise *F_ST_* analysis showed highly significant genetic differentiation between populations. The pairwise *F_ST_* values varied from 0.034 to 0.993, and the highest genetic differentiation was perceived between Population 3 (BK, Perak) and Population 4 (KB, Terengganu). The average *F_ST_* value in the present study was found to be 0.645.

Additionally, pairwise Tamura–Nei analysis showed highly significant genetic differentiation between haplotypes (Table 4). Tamura–Nei genetic distances measure the genetic divergence between populations based on DNA sequence data. The pairwise values below the diagonal in the matrix indicate the genetic distances between each pair of populations. The pairwise Tamura–Nei analysis value varied from 0% to 7.2%, and the highest genetic differentiation was observed in HAP_2 (*Batagur affinis edwardmolli* BP1). The average Tamura–Nei in the present haplotype study was 4.7%.

### 3.3. Population Demography History

Tajima’s D and Fu’s Fs tests were conducted to infer the demographic history and detect the past population growth of the Southern River terrapin. The Tajima’s D and Fu’s Fs test results are given in Table 5, including the associated simulated *p*-values. Tajima’s D values were negative except in Populations 3 and 4, indicating an excess of rare nucleotide site variants. On the other hand, Fu’s FS test results, based on the distribution of haplotypes, showed positive values for all the studied populations except for Population 1, signifying an excess of rare haplotypes.

In general, negative scores from the two tests (Tajima’s D and Fu’s FS) indicated an excess of rare mutations among the populations. In contrast, positive scores of two tests (Tajima’s D and Fu’s FS) suggest an excess of high-frequency variants, which can be consistent with population contraction or balancing selection. It is hard to find research that shows that the mitochondrial genome in natural populations has been changed by direct or indirect selection (via hitchhiking) [69].

We also considered the SSD and RI under the demographic expansion model for each population. As shown in Table 5, we found that all of the populations we looked at had non-significant SSD and RI except Population 4. This suggests that for Population 4 (i.e., *p*-value less than 0.05), the observed data did not fit the model of population expansion well and may be due to factors other than neutral evolution. Moreover, the data indicated departures from neutrality that may be due to factors such as demographic events or natural selection. For the other populations, however, the observed data was consistent with the model of population expansion and did not show evidence of departures from neutrality [10,47].

By examining the frequency distributions of pairwise differences across sequences, it is possible to gain insight into historical demographic expansions [70,71,72]. In the current study, the mismatch distribution plot was unimodal in Populations 1 and 3. On the other hand, Population 2 (Bukit Pinang, Kedah) has a bimodal distribution, and Population 4 (Kuala Berang, Terengganu) does not have an SNP (Table 2). This means there is no mismatch distribution for that population.

A unimodal peak was observed in both lineages of Population 1 (Kemaman, Terengganu) and Population 3 (Figure 3). These findings strengthen the evidence that both of these lineages underwent sudden expansion [73]. Bimodal distributions of population sizes in Bukit Pinang, Kedah, and total populations indicate diminishing population sizes or structured sizes, and a ragged distribution suggests that the lineage was widespread [10,38,74]. Moreover, the bimodal mismatch distribution showed the distribution of mutational fitness effects for mitogenomes [75] and often for individual genes, where most mutations are either neutral or lethal, with only a few having an intermediate effect [76].

### 3.4. Phylogenetic Relationship

A phylogenetic tree and a Bayesian analysis were undertaken to study the relationships and divergence of Southern River terrapins’ haplotypes, and the results supported the monophyly of *Batagur affinis* (Figure 4). The tree signified the relationship between the haplotypes from entirely different samples (Figure 4). The Bayesian evaluation presented identical tree topologies. Thus, phylogenetic analysis of haplotypes clearly supported the monophyly of *B. a. affinis* and *B. a. edwardmolli*. The phylogenetic analyses further classified the haplotypes of *B. affinis* into two groups (cluster 1 and cluster 2). However, cluster 1 was thought to consist only of the population from the Terengganu region (*B. a. edwardmolli*). Surprisingly, *B. a. edwardmolli* was found in a minority population in Kedah, Malaysia, where it was discovered together with HAP_2 (*B. a. edwardmolli* KE09) and related to HAP_4 (*B. a. edwardmolli* BP31). This indicates that the *B. a. affinis* individuals that are most common in Kedah and Perak are in cluster 2 (HAP_3, HAP_5, and HAP_6).

Haplotype network analysis was performed using the randomised Minimum Spanning Tree (MST) method. The median-joining haplotype network showed that HAP_4 is related to all the other haplotypes, indicating that it is the ancestral one, and the haplotype distribution was established on the map. In addition, a median-joining (MJ) network was created with six haplotypes based on 652 bp mtDNA D-loop sequences (Figure 5) to analyse and visualise the relationships between mtDNA D-loop sequences within the populations. The network demonstrations appeared as two separate clusters, with cluster 1 on the bottom and cluster 2 on the top. The HAP_1, HAP_2, and HAP_4 haplotypes were located in cluster 1, while the remaining haplotypes (namely HAP_3, HAP_5, and HAP_6) were found in cluster 2. In addition, the network showed a glass goblet profile constant with a population expansion in the past.

The map (Figure 2) shows sampling locations and corresponding mtDNA haplotype frequencies (absolute and relative frequencies are indicated by the numbers in parentheses and coloured fractions of the pie graphs, respectively) of *Batagur affinis* populations from Peninsular Malaysia. The examination was based on haplotypes of the mtDNA D-loop (ca. 652 bp) from a total of 120 individuals. HAP_2 was shown be found in the greatest number of localities compared to HAP_1, HAP_4, and HAP_6. The *B. a. affinis* from BK, Perak, and BP, Kedah, were both found to have the greatest variety of haplotypes.

## 4. Discussion

### 4.1. Population Diversity

This study examined the genetic variability of *Batagur affinis* populations in Peninsular Malaysia by analysing the mitochondrial D-loop. This is the first time that the mtDNA D-loop sequences of *B. a. affinis* have been investigated. The use of mitochondrial D-loop DNA sequences to study the genetic variation of *B. a. edwardmolli* has been previously described [31]. Meanwhile, other studies [8,77] used the mitochondrial cytochrome-b gene to investigate the genetic variation of *B. affinis*, but the results were conflicting due to the small number of samples. In consequence, we investigated the genetic variation of four different *B. affinis* populations found on Malaysia’s east coast (KE and KB, Terengganu) and west coast (BK, Perak, and BP, Kedah). Some haplotypes were shared between the regions, indicating common sources of colonists. The six haplotypes discovered in the D-loop area of the terrapins (compared to [31]) are attributable to the huge number of samples collected and the significant substitution rate observed in the D-loop region. According to [31], a total of 784 bp of mtDNA sequence yielded only one haplotype across all four groups and one ungrouped individual. This finding means that the D-loop region of the mitochondrial DNA genome could have a higher substitution rate than the rest of the mitogenome [14].

When compared to populations of other *Batagur* species, our Hd value was 0.214 on average. Similarly, an average Hd value of 0.2 was reported for Cambodian *Batagur affinis edwardmolli* (Hd value average = 0.2) [31], and, again, a higher Hd value average (0.405) for *Batagur borneoensis* was reported by [78]. This implies that the genetic variability in the BK of the Perak population was the highest in this region. In contrast, the lowest genetic variability was detected in the KB population of the Terengganu region. According to the results of the current study, a low Hd, a significant Hd test result, and low variability can indicate a population is rapidly growing demographically from a small but viable population size [79]. However, more research needs to be done to confirm these results and rule out other possibilities. For instance, additional studies could investigate the historical demographic trends of the population or use different genetic markers to verify the results. It is also important to consider other factors that could influence genetic variability, such as mutation rates, gene flow, and selection pressures. Ultimately, a more comprehensive understanding of the population’s genetic diversity and demographic history would be necessary to fully assess the implications of these findings.

The percentage of haplotypes that were similar reached 67.8% in this study. As previously stated, it has been demonstrated that there is a critical situation for the *Batagur affinis* population in Malaysia. The π value measured was low, with an average difference in nucleotide composition (k) of 0.3%. This low value indicates that a population bottleneck might occur because of hunting for consumption and retail, habitat loss, and inbreeding in the restricted habitat. All of these could be contributing to *B. affinis’* decline. Additionally, low π indicated minor haplotype differences [80]. Based on these regions, the population diversity was higher, with two populations of *B. affinis affinis* and three different haplotypes found in four other regions. In addition, the total number of populations was highly monomorphic (96%). A hypervariable region with high base substitution, the D-loop, allows for more accurate identification of individual diversity than the other regions [78]. Given that this region is well-known as a hypervariable in vertebrates, meaning it has a high base rate of mutation, whether substitution, deletion, or insertion, it is widely used to determine the level of intra-species diversity [81,82].

Similar biochemical processes cause polymorphisms and mutations, but the term “polymorphism” is used to avoid implying that any particular allele is normal or abnormal. Some of the consequences of inbreeding can also be explained by mutations [83,84]. As a result, the highest number of SNPs were detected in the population BP, Kedah (Table 2). SNPs are simply genetic variations that occur within a population and can occur naturally through mutations and genetic drift [85].

### 4.2. Population Structure

This study discovered significant intra-population and inter-population variations in the *Batagur affinis* populations. Variations within populations were found in all the selected *B. affinis* populations. This could be explained by one or more factors, such as small population sizes, previous bottleneck events, or the existence of physical barriers to gene flow between populations [86]. Most *Batagur affinis* populations have fixed haplotype differences and high *F_ST_* values. This means that there was not much gene flow between the populations, which could be because rivers or large distances separated the populations [87]. A similar result in *Dermochelys coriacea* in the United States of America [88] supports the high *F_ST_* values found in this study.

Based on DNA sequence data, Table 4 shows a summary of how far apart the four populations of *Batagur affinis* are genetically and how different they are from each other. The data can be used to infer patterns of gene flow, genetic diversity, and population structure among the populations, which can be important for the conservation and management of this species.

Nonetheless, a particular haplotype is distributed among several locations; for example, KE-KB, Terengganu, and BP, Kedah shared HAP_2. *Batagur affinis affinis* populations from Perak and Kedah shared mainly HAP_3. As a result, this would suggest that *B. a. affinis* had a historically extensive natural distribution in the area.

### 4.3. Population Demography History

To study the population history, neutrality tests were conducted using Tajima’s D [42] and Fu’s FS statistics [46]. These two tests calculate deviance from neutrality by assuming a continual population growth at mutation-drift equilibrium. A negative Tajima’s *D* value (as measured from KE, Terengganu, and BP, Kedah populations) shows an excess of low-frequency polymorphisms relative to expectation [89], indicating population size expansion due to positive selection [44]. A positive Tajima’s *D* (as estimated from the BK, Perak, and KB, Terengganu populations) indicates an excess of high-frequency variants, which can be consistent with population contraction or balancing selection [89,90]. In all populations, the Tajima’s D and Fu’s FS neutrality tests are statistically non-significant as the *p*-value is greater than 0.05, which means there is no evidence of deviation from neutrality in the population. It suggests that the observed genetic variation in the population is consistent with what would be expected under neutral evolution, and there is no reason to reject the null hypothesis of neutrality [91].

Meanwhile, a negative value for Fu’s *F_S_* indicates an excess of rare alleles in the population. This can be an indication of recent population expansion or a selective sweep, which can result in the loss of genetic diversity in the population. In contrast, a positive value indicates the opposite [89]. Since Fu’s FS neutrality test *p*-value for Population 4 is undetected, its significance remains unknown.

All the statistically non-significant tests were consistently compatible with the population’s being in drift-mutation equilibrium, as determined by Tajima’s *D* and Fu’s *F_S_* test statistics. Nonetheless, Fu’s simulations denoted that Fu’s FS was a more sensitive indicator of population expansion than Tajima’s D [46].

According to [47], the SSD and the RI were used to test the fit of genetic data to the spatial expansion model in population genetics of the *Batagur affinis* population in Malaysia. This was done to test the hypothesis that the experimental records fit the unexpected expansion model [48]. The SSD and RI values calculated were statistically non-significant, except for Population 4 (KB, Terengganu), indicating that the genetic data of the other populations are consistent with the spatial expansion model, while Population 4 may have undergone a different evolutionary process that has affected its genetic diversity. When the *p*-values of these tests are greater than 0.05, it suggests that the observed pattern of genetic diversity is consistent with the spatial expansion model and that there is no evidence of deviation from this model. However, the *p*-value for Population 4 is less than 0.05, which means that the observed pattern of genetic diversity is very different from the spatial expansion model [49]. It is important to note that further analysis is usually required to determine the specific evolutionary process that has caused the deviation from the spatial expansion model in Population 4.

### 4.4. Phylogenetic Relationship

Regarding population differentiation, the results of haplotype network analysis established that Population 2 (BP, Kedah) and Population 3 (BK, Perak) share the most prevalent haplotypes and exhibit only minor divergence. High gene flow between populations can slow down or prevent geographic differentiation. The populations from BK, Perak and KB, Terengganu exhibited the most differentiation, as the geographical distance between the west coast of Peninsular Malaysia and the east coast of Peninsular Malaysia, populated by these terrapins, divides the two subspecies. It is possible that *Batagur affinis*, commonly known as the Southern River terrapin, is present in the waterways within the territory between the east and west coasts of Peninsular Malaysia.

Accordingly, the phylogenetic interpretation of the haplotype network revealed a “glass goblet-like” topology with a large proportion of singletons. This is typically interpreted as a sign of a growing population, particularly from a small number of founders following a population bottleneck [92]. Here, HAP_4 is the possible ancestral haplotype, while HAP_3, HAP_5, and HAP_6 may have formed one cluster. Without much gene flow, the clade distribution may be useful in determining the genetic structure of a local population.

### 4.5. The New Distribution Range of Batagur affinis edwardmolli Is a Novelty

The topologies of our Bayesian phylogenetic tree were broadly concordant and showed that HAP_1, HAP_2, and HAP_4 could be grouped as cluster 1 (*Batagur affinis edwardmolli*). On the other hand, HAP_3, HAP_5, and HAP_6 could be grouped as cluster 2 (*B. a. affinis*). The genetic distance between HAP_2 *B. a. edwardmolli* KE09 and HAP_4 *B. a. edwardmolli* BP31 distances were 0.2% to 0.3%. Therefore, with a significant bootstrap confidence level (>50%) in Appendix A, the new distribution range of *B. a. edwardmolli* populations in Malaysia was revealed. This subspecies similarity was discovered in Kedah, on the west coast and north of Peninsular Malaysia (Figure 6). According to [8], *B. a. edwardmolli* was distributed only along the east coast of Peninsular Malaysia and adjacent Thailand. In contrast to [8], we currently find *B. a. affinis* and *B. a. edwardmolli* in the northern region of Peninsular Malaysia (Kedah). In addition, Population 2 (*B. affinis* from Bukit Pinang, Kedah) shared a HAP_2 with Population 1 and Population 4 from Terengganu, *B. a. edwardmolli*. Based on the history recorded, Southern Thailand has two subspecies of *B. affinis* [8]. Geographically, to the north, the Kedah border shares an international border with Southern Thailand. Moreover, [2] stated that *B. a. affinis* could be found in the southernmost region of Thailand. Furthermore, *B. a. edwardmolli* was previously recorded found in the Songkhla region of Thailand’s neighbouring, southernmost eastern peninsula. A Thailand conservation centre for *B. affinis* is reported to be located on the Klong Langu River in Satun Province (Southern Thailand). Therefore, the population of Southern Thailand is expected to migrate to Kedah, Malaysia. This could be due to the nature of the disaster (e.g., flooding) and the high threats faced by *B. affinis* in Thailand, which forced the population to migrate outside of Thailand and into Kedah, Malaysia, in search of better resources or living conditions. The author of [2] reported the population as extinct in the wild in Thailand, and the wild population experienced a severe decline long ago [93]. Moreover, based on this study, the analysis of bimodal mismatch distribution indicated potentially that Population 2 from BP, Kedah, is a high-strength and high-survival rate population with fitness effects of mutations that support the species’ ability for long-distance migration. However, further studies on Thailand’s populations are needed to test this hypothesis.

### 4.6. Conservation Impact and Recommendation

According to [93,94], the long life expectancy of *Batagur affinis* indicates that the wild populations of this species have been experiencing a severe decline for a long time. The studies also stated that the few survivors are mainly related at the full-sib and half-sib levels. In view of this issue, they proposed a conservation strategy that preserves as much of the current genetic diversity as possible in order to avoid inbreeding. Thus, they also advocated for selective breeding and the avoidance of certain mating combinations.

Furthermore, like the Cambodian population, the Malaysian population has a minimal population size [95]. Finally, given the severity of the bottleneck and the population’s strong kinship divisions, inbreeding is expected, but further study should be carried out to examine this. For instance, a study could be done to find out how often and what the possible effects of inbreeding are in a certain population, such as a remote tribe living in an isolated area. Alternatively, researchers could analyse the genetic diversity of a specific animal population that inhabits a small and isolated habitat to assess the degree of inbreeding and potential threats to its survival. Ex-situ conservation actions to minimise inbreeding within the *Batagur affinis* populations are thus life-threatening for the species’ conservation, as inbreeding depressions can have speedy and exponentially growing deleterious impacts on such small populations [22,23].

Given the information on the population’s genetic and demographic impoverishment, retaining contemporary genetic diversity should be the primary management goal. As a result, a fine-scale valuation of population structure and pedigree examination are required to choose ideal individuals to ensure colony management and/or reintroduction efforts [31]. Likewise, although the number of conservation genomic studies has been increasing (see examples in [96,97]), it would be necessary to start studies with genomic approaches in order to continue promoting and developing genomic procedures in the conservation of non-model and non-commercial species [98].

Our research findings can begin to fill a knowledge gap by shedding light on genomic variation, population structure, and demographic parameters in a relict population of one of the most critically endangered turtle species. Most of the world’s turtle species are endangered. Hence, we used genetic information to inform the relevant decision makers on proper colony management and contribute to the conservation of *Batagur affinis* genetic diversity while bridging the gap between in-situ and ex-situ conservation. Moreover, given that 70% of the world’s most endangered turtles live in Asia and have demographic histories comparable to those of *B. affinis,* our genomic technique can be extended to a wide range of turtle species in the long term.

Based on the situations described above, we suggest having an exchange breeding programme (same subspecies) among DWNP Kedah, Zoo Negara, and Zoo Melaka to reduce the potential for inbreeding in the BP Kedah population. Meanwhile, an exchange programme between the KE and KB populations is required to enhance genetic variance among the KB (Terengganu) populations. Additionally, we proposed that additional research should be conducted, such as more exploration to find new wild population sites and conducting a whole-genome sequencing study to reveal more important genetic matters crucial for conservation strategy. At this time, the ranching conservation programme should be maintained, while any sand mining should be stopped immediately. Additionally, habitat enrichment is required in at least four population regions to protect successful hatchlings from hunters and natural predators during the egg and hatching seasons.

## 5. Conclusions

Our research showed low genetic diversity and significant genetic differences across the terrapin populations and revealed a novel distribution range for *Batagur affinis edwardmolli* populations in Malaysia. This information is helpful for conservation programmes aimed at sustaining the genetic stock.

## Figures and Tables

**Figure 1 biology-12-00520-f001:**
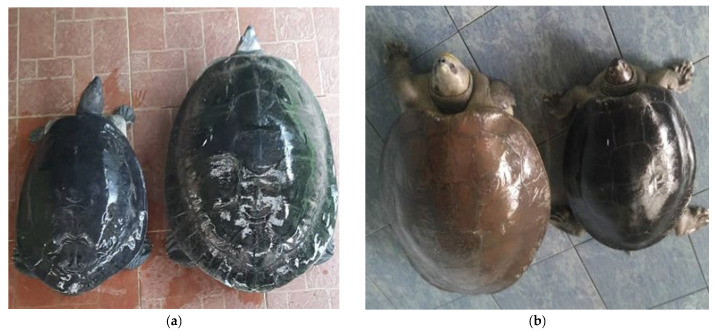
(**a**) Male *Batagur affinis affinis* (**left**): The head is noticeably shorter, with a blunt snout and a shorter distance from nostril to the eye; the head is jet black or extremely dark grey, never brownish; and the iris is pristine white throughout the mating season. Female *B. a. affinis* (**right**): The head is short and has a blunt snout; the space between the nostrils and the eyes is very short; the head is dark grey or brownish, the jaws are dirty yellow or light brown, and the other soft parts are dark grey or brownish. (**b**) Female *B. a. edwardmolli* (**left**): The head is elongated, with a pointed, upturned snout; the head is greyish to brownish, with pale grey to silvery blotches in the temporal and parietal areas, and the jaws are brown. Male *B. a. edwardmolli* (**right**): Head and soft parts chocolate brown to almost black; light-colored individuals with brown and never grey skin; lip margins orange; iris golden or bright yellow. dark brown to black carapace [6].

**Figure 2 biology-12-00520-f002:**
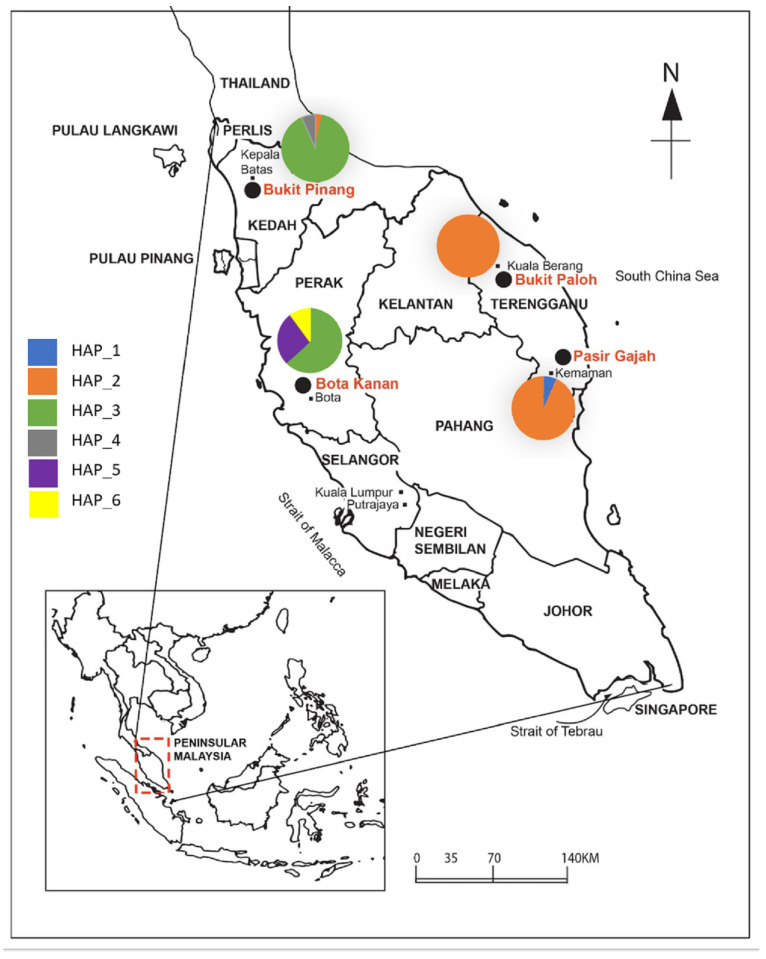
Distribution of haplotypes in sampling locations (see Results section). Samples collected from four study sites in Peninsular Malaysia (Bukit Pinang (BP), Kedah; Bota Kanan (BK), Perak; Bukit Paloh, Kuala Berang (KB), Terengganu; Pasir Gajah, Kemaman (KE), Terengganu).

**Figure 3 biology-12-00520-f003:**
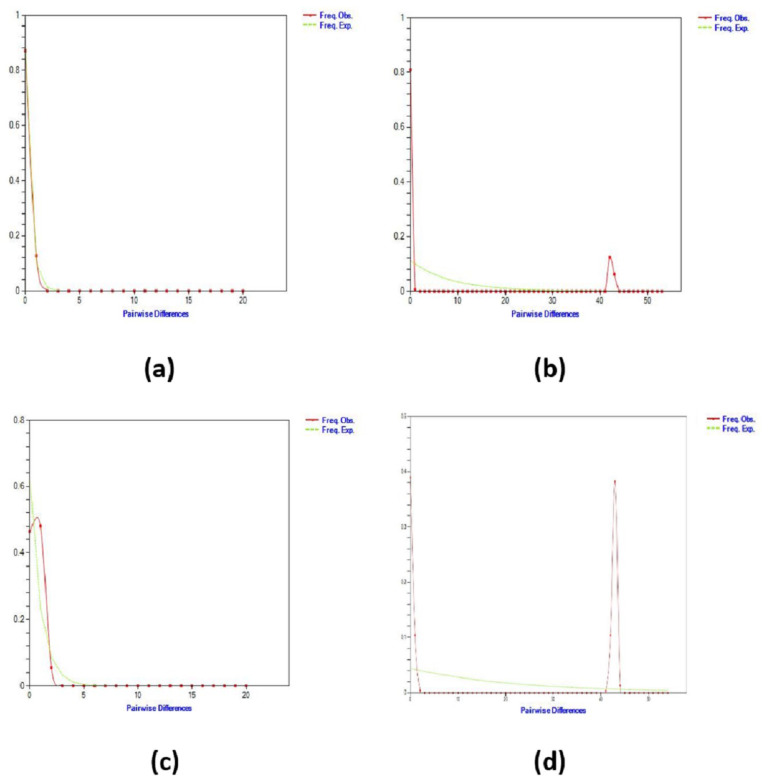
Mismatch distribution graphs of *Batagur affinis* for the (**a**) Pasir Gajah, Kemaman (KE), Terengganu (**b**) Bukit Pinang (BP), Kedah (**c**) Bota Kanan (BK), Perak, and (**d**) total populations.

**Figure 4 biology-12-00520-f004:**
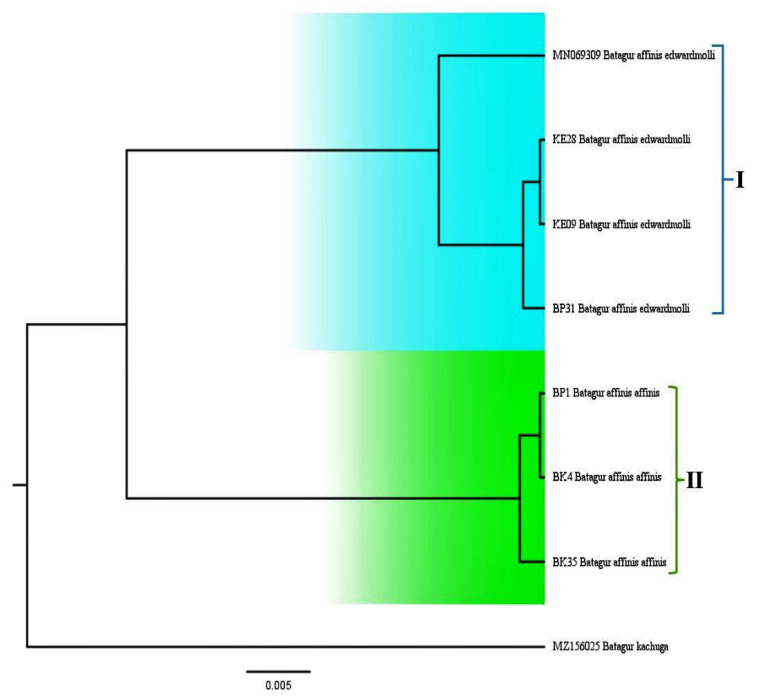
Phylogenetic tree showing the relationship, generic, and species positions based on Bayesian analysis of the mtDNA D-loop (652 bp) of *Batagur affinis*. The phylogenetic tree also shows the relationship between *B. a. edwardmolli’s* D-loop haplotypes and *Batagur kachuga* as the outgroup.

**Figure 5 biology-12-00520-f005:**
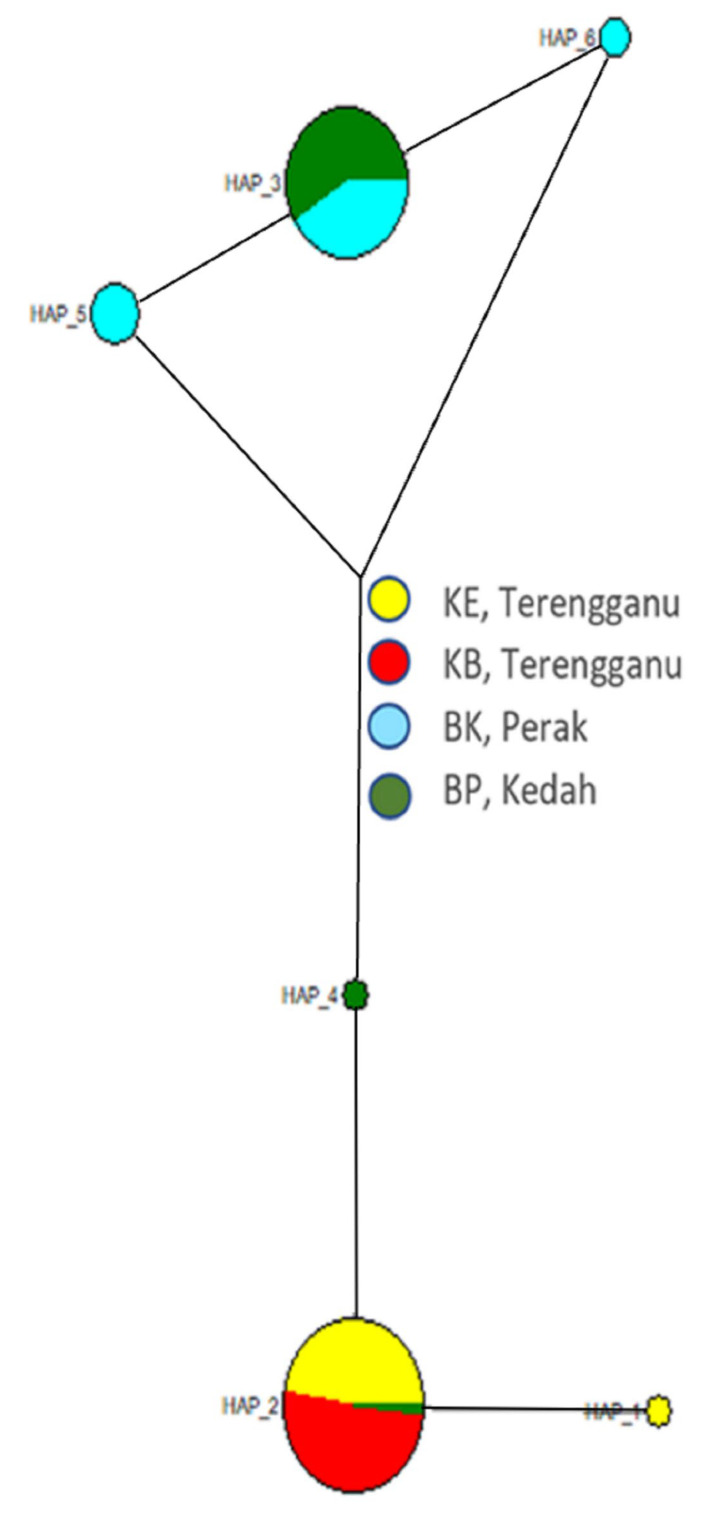
The Median-joining (MJ) network is a haplotype network with six haplotypes identified from 120 sequences of D-loop fragments. Pie charts are proportional to the frequency of haplotype sequences, and branches are scaled according to the number of mutation steps, shown with lines above the branches. The most frequent haplotype (HAP_2) is found in populations from Pasir Gajah, Kemaman (KE) Terengganu, Bukit Paloh, Kuala Berang (KB), Terengganu, and Bukit Pinang (BP), Kedah. HAP_3 is shared by populations from Bota Kanan (BK), Perak, and Bukit Pinang (BP), Kedah.

**Figure 6 biology-12-00520-f006:**
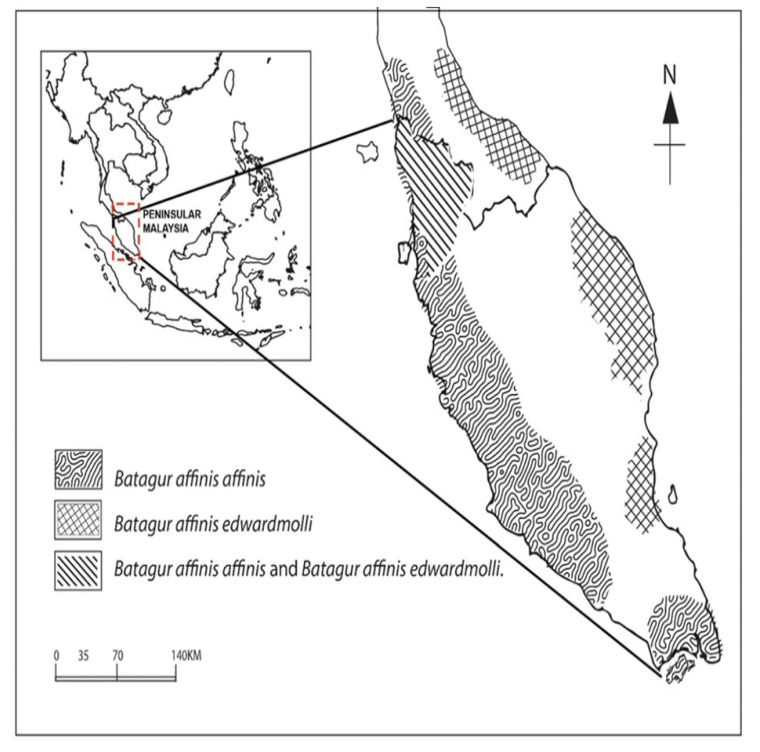
The distribution range of *Batagur affinis affinis* and *Batagur affinis edwardmolli* in Peninsular Malaysia.

**Table 1 biology-12-00520-t001:** Summary of SNPs of six observed mitochondrial DNA D-loop haplotypes of *Batagur affinis*. We designated the corresponding number to each SNPs with the first base of the mtDNA control region [65]. Dots indicate identity to the KE28 haplotype sequence.

Haplotype	Single Nucleotide Polymorphisms (SNPs)																																																	
												1	1	1	1	1	1	1	1	1	1		1	1	1	2	2	2	2	2	2	2		3	3	3	3	3	3	3	3	3	4		4	4	5	5	5	5
			1	1	2	3	4	4	7	7		1	2	2	3	3	4	4	4	5	5		5	5	9	0	1	1	1	7	9	9		3	5	5	5	6	6	7	8	8	2		3	4	1	4	6	7
	1	2	1	9	3	4	1	7	2	3		9	5	8	5	7	2	7	8	2	4		5	7	3	9	1	7	9	5	5	6		7	4	5	6	0	1	8	0	5	7		7	2	6	4	4	3
*Batagur affinis edwardmolli* KE28	C	A	T	A	A	G	G	G	A	T		A	A	T	G	G	T	G	G	A	C		T	G	G	C	T	T	T	A	C	T		G	C	C	T	C	T	A	T	C	G		A	A	C	T	C	T
*Batagur affinis edwardmolli* KE09	.	.	.	.	C	.	.	.	.	.		.	.	.	.	.	.	.	.	.	.		.	.	.	.	.	.	.	.	.	.		.	.	.	.	.	.	.	.	.	.		.	.	.	.	.	.
*Batagur affinis edwardmolli* BP31	.	.	.	.	C	A	.	.	.	.		.	.	.	.	.	.	.	.	.	.		.	.	.	.	.	.	.	.	.	.		.	.	.	.	.	.	.	.	.	.		.	.	.	.	.	.
*Batagur affinis affinis* BP1	A	G	G	G	C	A	A	A	G	C		G	G	C	A	A	C	A	A	G	T		C	A	A	T	C	C	C	G	T	C		A	T	T	C	T	C	G	C	T	A		G	G	T	C	G	C
*Batagur affinis affinis* BK4	.	.	G	G	C	A	A	A	.	C		G	G	C	A	A	C	A	A	G	T		C	A	A	T	C	C	C	G	T	C		A	T	T	C	T	C	G	C	T	A		G	G	T	C	G	C
*Batagur affinis affinis* BK35	A	G	G	G	C	A	A	A	G	.		G	G	C	A	A	C	A	A	G	T		C	A	A	T	C	C	C	G	T	C		A	T	T	C	T	C	G	C	T	A		G	G	T	C	G	C

**Table 2 biology-12-00520-t002:** Sample size, number of haplotypes, number of single-nucleotide polymorphisms, haplotype diversity, and nucleotide diversity of different sites of *Batagur affinis*.

Population (ID) *	No. of Haplotypes	Sample Size	SNPs	Haplotype Diversity (Hd)	Nucleotide Diversity (π)
1 (KE)	2	30	1	0.1287	0.0002
2 (BP)	3	30	43	0.1908	0.0123
3 (BK)	3	30	2	0.5356	0.0009
4 (KB)	1	30	0	0.0000	0.0000

* 1: Pasir Gajah, Kemaman (KE), Terengganu; 2: Bukit Pinang (BP), Kedah; 3: Bota Kanan (BK), Perak; 4: Bukit Paloh, Kuala Berang (KB), Terengganu.

**Table 3 biology-12-00520-t003:** Hierarchical analysis of molecular variance (AMOVA) and spatial analysis of molecular variance (SAMOVA) among populations of *Batagur affinis*.

Source of Variation	*df*	Sum of Squares	Variance Components	% Total Variance
**AMOVA**				
Among populations within groups	3	1162.850	12.885	92.29
Within populations	116	124.800	1.076	7.71
Total	119	1287.650	13.960	100
**SAMOVA, 2 groups**				
Among groups	1	1167.241	19.241	94.39
Among populations within groups	2	6.424	0.071	0.35
Within populations	117	125.641	1.074	5.27
Total	120	1299.306	1299.306	100

**Table 4 biology-12-00520-t004:** Pairwise Tamura–Nei genetic distances (below the diagonal); Pairwise Fixation Index (*F_ST_*) and the probability (significant Chi-square test in bold value; *p*-value < 0.05) for population differentiation based on 110 permutations of the sequence dataset (above the diagonal) among four populations of *Batagur affinis*.

Population (ID) *	1 (KE)	2 (BP)	3 (BK)	4 (KB)
1 (KE)	-	**0.896**	**0.991**	0.034
2 (BP)	0.065	-	0.063	**0.898**
3 (BK)	0.072	0.008	-	**0.993**
4 (KB)	0.000	0.065	0.072	-

* 1: Pasir Gajah, Kemaman (KE), Terengganu; 2: Bukit Pinang (BP), Kedah; 3: Bota Kanan (BK), Perak; 4: Bukit Paloh, Kuala Berang (KB), Terengganu.

**Table 5 biology-12-00520-t005:** Tajima’s *D*, Fu’s *Fs* neutrality tests, Sum of Squares Deviation (SSD) and Raggedness Index (RI) with related *p*-values.

Population (ID) *	Tajima’s *D* (*p*)		Fu’s *F_S_* (*p*)		SSD (*p*)		RI (*p*)	
1 (KE)	−0.76373	0.229	−0.43926	0.155	0.04026	0.090	0.56792	0.390
2 (BP)	−1.01094	0.154	13.82783	0.999	0.03459	0.060	0.67051	0.690
3 (BK)	0.34880	0.698	0.39996	0.555	0.02202	0.080	0.18417	0.060
4 (KB)	0.00000	1.000	0.00000	NA	0.00000	0.000	0.00000	0.000

* 1: Pasir Gajah, Kemaman (KE), Terengganu; 2: Bukit Pinang (BP), Kedah; 3: Bota Kanan (BK), Perak; 4: Bukit Paloh, Kuala Berang (KB), Terengganu.

## Data Availability

Sequences generated in the study have been deposited in GenBank sequence database with accession number in the text. Raw data can also be provided upon request.

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
