# Peer review of "Conservation Genetics of the Critically Endangered Southern River Terrapin (Batagur affinis) in Malaysia: Genetic Diversity and Novel Subspecies Distribution Ranges"

_biology, 2023, doi:10.3390/biology12040520_

Round 1

Reviewer 1 Report

biology-2194897 - Genetic Variation of the Southern River Terrapin Populations: A New Distribution Range and Conservation Genetics in Malaysia

Mohd Hairul Mohd Salleh and Yuzine Esa

Letter to the Authors

I am recommending that this paper be published after major revision.

Turtle species in SE Asia are generally threatened. Any research contributing to our knowledge of them is to be encouraged, particularly research on population genetics, which is vital in framing conservation strategies for small and scattered populations and can also identify hidden genetic variation. I thus would like to see this contribution published, but I think that you need to re-write significant portions of it, and also think a bit more about your interpretations of some of your statistics.

The accompanying Word file has a lot of marginal notations, some containing suggestions for improving the writing, and others criticizing the presentation of the statistical results and their discussion. My detailed criticisms are to be found there. However, a few general comments are below.

The figures are blurry. This makes them hard to interpret. I don’t know if this is what they were like when you submitted but it must be fixed.

There is no need to write Batagur affinis ssp. when you are referring to the species as a whole – just the species name will do.

The quality of English is uneven – some sections (Introduction, Methods and Material) are quite clear, but in others I had a hard time deciding what was being said (Results and particularly Discussion). I have made suggestions throughout the ms of ways that a native English speaker would put things, when I thought that I could puzzle out what was meant, and I hope that these are helpful. However, in the Discussion, I more or less gave up – it was very difficult to follow what you were saying. Rather than going into detail here, I refer you to the numerous marginal comments in the attached Word file. I realize that writing in a second language is difficult and the problem is compounded when you are writing a scientific paper, but I honestly feel that the Discussion has to be re-written. A professional translating service may be of aid here. I’m stressing this because I think that your findings are important and should be published, but we have to be able to make sense of what you have to say about them.

I had some problems with the writing in the Results section as well, and would recommend a thorough revision of this portion also. Again, my marginal comments will, I hope, help with this.

While your use of statistics is appropriate, you based conclusions on non-significant p-values. If your parameter estimates aren’t statistically significant, you can’t base any conclusions upon them – these must be re-considered. There are comments on this in the accompanying Word file.

I was puzzled as to why you chose Dermochelys coriacea as your outgroup. It’s phylogenetically quite distant from Batagur affinis. Couldn’t you find comparable sequences for another species of Batagur?

I was also puzzled to see two subspecies sharing part of their range (Fig. 5). Subspecies are generally considered to be distinct geographic variants of a species, with allopatric distributions. Batagur a. affinis and B. a. edwardmolli could presumable interbreed where their distributions overlap, and lose their subspecific identities. Also, it appears that this region of geographic overlap corresponds largely to the area covered by the state of Kedah and in fact seems to be limited by the actual borders of the state. Some discussion of this is indicated.

I’m sorry if this review appears to be unremittingly negative, but given the conservation status of so much of Southeast Asia’s turtle fauna, studies of species’ population genetics are very important. As I said above, I think that this paper should be published, but in its present form the scientific community would have a hard time elucidating its findings.

Author Response

REVIEWER’S COMMENTS

Thank you very much for reviewing my manuscript. I also greatly appreciate the reviewers' insights and suggestions to improve this paper. I have carried
out the analysis that the reviewers suggested and revised the manuscript accordingly.
2. I have attached the manuscript, along with point-by-point highlighted changes made to address the reviewer's concerns. I hope that you find our responses satisfactory.

Thank you.

Reviewer 2 Report

In this research, the authors applied a conservation genetics approach to Southern River Terrapin with different haplotypes from D-loop polymorphisms. In principle, it seems to be an interesting article. Nevertheless, I have found several issues in the Results section that made it a little difficult for me to review the Discussion. These issues must be addressed. Additionally, I found some typos that should be corrected. The pending issues that need to be addressed before the manuscript can be considered for publication are in the attached pdf file.

Author Response

(The authors gave the same response as above.)

Round 2

Reviewer 1 Report

A Word file of the annotated  ms accompanies this review.

I am again recommending that this paper be published, but with major revisions. The importance of understanding the population genetics of the beleaguered SE Asian turtle species cannot be stressed too much, but analysis of data must be solid.

In my last letter to you, I listed a series of criticisms that I thought needed to be addressed. Your letter of response did not deal with these, and it slowed the process of re-evaluation down considerably. While suggestions regarding grammar and word usage don’t require point-by-point responses – you highlighted everything that you changed in this regard anyway, and that was helpful – questions and criticisms concerning methods and results, whether in a letter like this or in marginal comments, do require responses, in some depth, in an accompanying letter.

I’ve gone through the revised ms and made some further suggestions regarding word use etc. My apologies for not catching these the first time through.

Again, the writing was generally clear in the Introduction, the Methods, and the Results. However, I still have a lot of trouble with the Discussion, both with the writing and with the interpretation of the population genetics statistics (Tajima's D, Fu’s FS, SSD and Raggedness). There are marginal notes concerning the writing – again, I apologize for things that I missed last time – but again, I would encourage you to get a native English speaker, one who you can talk to in person, to run through the text of the Discussion and make suggestions about the writing.

You put a lot of weight on the population genetics statistics, as given in Table 5 - Tajima's D, Fu’s FS, SSD and Raggedness  - but almost all of these estimates have probabilities greater than 0.05 and are thus not significant – the null hypotheses can’t be rejected. In light of this, any discussion based upon these statistics must be restricted to this finding – nothing further can be said, and conclusions based upon the values of these statistics are not valid. In addition, these are simultaneous estimates, and a stepwise adjustment of the acceptance levels must be made for each statistic. I brought this up in my previous review but you haven’t responded effectively to these points. You must either tell us why an estimate of a statistic with a probability of > 0.05 can be interpreted as if it was significant, or re-interpret your findings to accommodate the lack of significance. This is a major point and must be addressed in your letter of response.

On Fig. 6, you again depict an area within which Batagur a. affinis and B. a. edwardmolli are apparently sympatric. I asked in my previous review how two subspecies could share a geographic area – subspecies are generally allopatric. This question was not answered.

There are some other question raised in marginal comments in the Discussion. I won’t get into them here but they also need to be answered.

Again, I’m sorry to sound so harsh, but I realize what a lot of work you’ve done, and I think that it’s important that it be interpreted properly. I want to see this published, and encourage you to answer my questions, in detail, or re-write the relevant text.

Author Response

Thank you for sharing your thoughts. Even though sometimes you are so harsh, we are practising answering in a polite way. You are pretty awesome for also editing my grammatical errors. Thank you so much for your efforts to ensure this manuscript looks perfect.

Reviewer 2 Report

Journal: Biology (ISSN 2079-7737)
Manuscript ID: biology-2194897
Type: Article
Section: Conservation Biology and Biodiversity
Special Issue: Aquatic Biodiversity and Ecosystem Multifunctionality in Response to Environmental Changes

The authors have implemented most of the suggested changes recommended in the first revision in this new manuscript version, adequately arguing most of the issues raised. As a result, the quality of the manuscript has improved adequately. Nevertheless, I observed some typos, errors (p-value = 2.081 in table 5), confusing sentences, blurring images (Figs. 2-3) and some doubts about the interpretation of the Results in Discussion section. The pending issues that need to be addressed before the manuscript can be considered for publication are listed below in two sections: (1) Content issues and (2) Formatting issues.

Content issues

Lines 26-27: If you agree, please change "We also created neutrality tests, namely the Tajima's D test and Fu's Fs test were used to evaluate the signatures of recent historical demographic events." TO

"Tajima's D test and Fu's Fs neutrality tests were performed to evaluate the signatures of recent historical demographic events."

These tests were not created in this research.

Lines 45-46:  The phenotypic differences observed in Figure 1 between the two subspecies seem quite evident. Are they so different? I recommend including in the figure caption some more sentences explaining some differences that are characteristic between the two (e.g., different coloration). The phenotypic differences observed in Figure 1 between the two subspecies seem apparent. Are they really that different? I recommend including in the figure caption a further sentence explaining some characteristic differences between the two (e.g., different coloration).

Lines 68-70:  According to our debate about it within the first-round review, I think that these sentences must be changed for something like this:
"Also, mitochondrial DNA polymorphisms have remained relevant to the study population structure and intraspecific variation [16-21]." I repeat, either I need clarification, or there is no upward trend in using mitochondrial markers.

Line 251: Please change "more significant" to "significant".

Lines 263-265: Please explain the table in a clearer way. What are the above values? FST or the probability (Chi-square test) values?

Line 278 (Table 5): Please, remove the last row (Total). This is mandatory for me. Either I need clarification, or it would not make much sense to add p-values, for example (p-value = 2.081).

Lines 371-374: Maybe I need a clarification. I did not understand this interpretation of the results. Low genetic diversity, a non significant p-value...a signal of a small but viable population in demographic expansion?

Line 392: "the highest SNPs were detected in the population BP". What is the meaning of this sentence? Sorry, but I did not understand.

Lines 394-395: Please remove the last sentence, it is redundant (the species is Critically Endangered according to IUCN's Red List).

Lines 399-400: "All populations except Population 1 have found significant differences in their mtDNA, which shows that they are different subspecies." I understood that populations 1 (KE) and 4 (KB) with haplotypes 1-2 are B.a.edwardmolli . But with this sentence one reader could understand another thing. If you agree, I would recommend rewrite this sentence. Is there a correspondence between haplotypes assigned to different subspecies and observed phenotypes observed in Figure 1?

Lines 409-410: I did not understand well this sentence and the meaning of "mixed subspecies" in the context of this research. Could you explain, please?

Line 441: "raggedness"?"Further, non-significant raggedness values on Population 4 also indicate population expansion". I do not quite understand this conclusion from the results obtained.

Lines 443-446: Maybe I am wrong, but from my view, these sentences are a bit contradictory between them.

Lines 504-505: What further study? Please add some example at the end of this sentence.

Lines 514-515: If there are no genomic approaches of this species, I think it would be more appropriate to write this: "it would be necessary to start studies with genomic approaches".

Line 532: "ranching conservation programme"? Sorry, but I did not understand well.

Line 537: Please change "demonstrated" to "showed".

Formatting issues

Line 104: Please change "(at a 1:3 ratio)" to "(1:3 ratio)"

Line 188-189: The link did not work.

Line 211: Please change "Batagur affnis" to "Batagur affinis"

Author Response

Thank you for sharing your thoughts. I like every single one of your comments, and we are practising answering in a polite way. You are pretty awesome for extracting every question clearly. Thank you so much for your efforts to ensure this manuscript looks perfect.

Round 3

Reviewer 1 Report

I have attached a Word file of your revised ms, with a few marginal comments and emendations of the text (mostly correction of typos). There is also a Word file of your covering letter with responses to your responses. This site only allows me to upload one file to accompany this letter, so I will have to get the editors to forward this second file to you  some other way. There are a couple of relatively minor issues discussed there that I think still need attention, but these should not be difficult to fix - I've made suggestions and tried to explain why I flagged these matters in the first place. On the whole, however, I found this revision to be easy to read and understand.

One question that I had that you did not seem to address - how can you have two subspecies occupying the same geographic area?

This is interesting work and very worthwhile. I hope that it helps efforts to conserve this species, and more broadly the turtle fauna of SE Asia, and I look forward to seeing it published soon.

Author Response

Thank you for your kind consideration of this manuscript. I have no problem going for round four after this. The manuscript gets much better round by round. Thank you.

Reviewer 2 Report

Journal: Biology (ISSN 2079-7737)
Manuscript ID: biology-2194897
Type: Article
Title: Genetic Variation of the Southern River Terrapin Populations: A New Distribution Range and Conservation Genetics in Malaysia
Section: Conservation Biology and Biodiversity
Special Issue: Aquatic Biodiversity and Ecosystem Multifunctionality in Response to Environmental Changes

In this new manuscript version, the authors have implemented almost all the suggested changes recommended in the previous revision, arguing the issues raised. As a result, the quality of the manuscript has improved adequately. Nevertheless, I observed some issues in the Results and Discussion sections and typos that must be corrected. The values in the main text with those in Table 4 do not match. Please check them. If well, most are minor things; correcting them is imperative and mandatory previous to the publication of this interesting research. I have attached a pdf file with the content and formatting issues.

Author Response

Thank you for your time in reviewing this manuscript. Round by round, the manuscript became much improved and clearer. I am so glad to have you. Thank you.
